# Chewing Revenge or Becoming Socially Desirable? Anger Rumination in Refugees and Immigrants Experiencing Racial Hostility: Latin-Americans in Spain

**DOI:** 10.3390/bs12060180

**Published:** 2022-06-07

**Authors:** María José da Silva Rebelo, Mercedes Fernández, Carmen Meneses-Falcón

**Affiliations:** 1University Institute of Studies on Migration, Comillas Pontifical University, 28015 Madrid, Spain; mariarebelo.ssps@gmail.com; 2Faculty of Humanities and Social Sciences, Comillas Pontifical University, 28015 Madrid, Spain; cmeneses@comillas.edu

**Keywords:** hostility, anger rumination, refugees, anger, social isolation, immigrants, Hispanic or Latino

## Abstract

This paper explores how real scenarios of racial hostility and discrimination trigger anger rumination tendencies in refugees, asylum seekers and immigrants (hereafter RASI). Undergoing discrimination often leads to the development of negative thoughts and behaviors, and to a loss of meaning and self-worth. This could make young RASI particularly vulnerable to being recruited and exploited by extremist groups as they search for identity. We developed a picture-elicitation instrument (the PEI) to provide professionals with a tool that could identify groups of RASI according to their reactions to discrimination scenarios and explore how racial hostility might influence withdrawal levels. The tool was applied with the Anger Rumination Scale (ARS_19) to 509 RASI of Latin American origin living in Spain. Four categories were identified, according to how RASI processed anger when observing discrimination scenarios: “Social desirability”, “Chewing”, “Grudge”, and “Vengeful”. Further analyses showed that the youngest (18–29) fell under the “Grudge” and “Vengeful” categories and revealed more despair and social isolation. This study makes a positive contribution by being the first to investigate the problem of anger rumination in RASI undergoing racial hostility. Moreover, it equips professionals with two tools that, once validated, may help plan and implement strategies to reduce the impact of hostility on both RASI and their host societies.

## 1. Introduction

The deteriorating effects of racial hostility on the general wellbeing of refugees, asylum seekers, and immigrants (hereafter RASI) are widely documented in the literature [1,2,3]; still, anti-migration movements, right-wing parties, and populist views are rising fast in Western countries [4]. Moreover, ethnic discrimination is continuing in social media and other media coverage [5], appearing as something normalized in European and North American political and social environments. The relevant literature shows [6,7] that media coverage of migration issues has a privileged place in shaping policies and public opinions in general, provoking a wave of dehumanization of RASI by their host societies [8].

This political and social anti-migration phenomenon, however, will not deter millions of people who are living in despair due to war, torture, persecution, or extreme poverty, from risking their lives in search of a better future for them and their families [9,10]. Regardless of their status as refugees, asylum seekers, or economic immigrants, the mistreatment these people endure before and after arriving in a host country has serious negative consequences for their mental health conditions [11].

The present work was conducted in Spain, where approximately 560,000 foreign people arrived during 2018 alone, a number that is comparable only to the number of arrivals in 2008. In 2019, this number continued to increase, with over 660,000 foreign nationals reaching Spain [12]. Latin American and Caribbean RASI reaching Spain have also risen sharply from the first semester of 2016 onwards. According to the same source, the number of Latin American women migrating to Spain was significantly higher than the number of men. This was particularly relevant in regard to people migrating from Honduras (70%), Nicaragua (71%), and Paraguay (69%), but it was also evident for people arriving from El Salvador (58%), Peru (56%), and Colombia (58%), among others. Human Rights Watch highlights that Spanish society used to be more tolerant of RASI than other European nations; nevertheless, hostile treatment and racial discrimination toward foreign nationals are rapidly increasing in this country [13].

In Spain, as in many other Western nations, RASI often struggle for a long time to overcome social and legal barriers. Factors such as labor exploitation, difficulties in renting housing, barriers in access to health care, along with experiences of racial hostility and discrimination, take a toll on this population’s wellbeing [14,15]. Moreover, these adverse factors often seriously compromise their personal and social identities, as well as their integration processes in their host societies [16,17,18].

### 1.1. Discrimination and Anger Rumination in RASI

In this article, we examine the extent to which racial hostility and discrimination experienced or perceived in host countries may influence anger rumination levels in RASI. Existing research has documented that anger rumination is correlated with difficulty in controlling negative thoughts and behaviors, and with high levels of frustration and irritation in the presence of anger-evoking scenarios [19,20]. We could not find studies in the literature that explored levels of anger rumination in RASI undergoing discrimination; nor could we find studies on how this discrimination might lead to negative coping mechanisms that feed into a vicious circle of hostility in Western countries.

Since this issue carries real-world implications for host societies, we built a new tool, the Picture-Elicitation Instrument (PEI), and used it along with the Anger Rumination Scale (ARS_19) to explore the presence of anger rumination in RASI living in Spain. Based on the results, we propose some suggestions for helping professionals and academics working with or conducting research with this population.

According to various studies, being treated in a hostile manner leads to deep feelings of sadness, powerlessness, isolation, or, in some extreme situations, strong feelings of anger or even aggression in those affected [21,22,23]. Moreover, feeling discriminated against, stereotyped, or mistreated on the basis of nationality, culture, or religion often triggers negative reactions or behaviors, such as anger or frustration, that may be harmful to both the stereotyped group and the host society in general [24]. This situation may have the potential to generate or further reinforce a vicious circle of hostility and anger on both sides; therefore, it is important to assess both the way RASI’s wellbeing is affected by Western societies’ hostile treatment, and how RASI cognitively process strong emotions that are activated by such mistreatment.

“Anger” is the most frequent emotional reaction triggered in those who experience racial hostility and discrimination. It consists of a strong emotions that result from feeling deliberately mistreated by somebody [25]. “Anger rumination” involves the way anger is cognitively processed [26], so it is related to the intensity and length of anger experiences; it may result from anger-triggering events that happen not just to oneself but to meaningful others as well [27]. It exacerbates feelings of irritation and frustration in anger-triggering scenarios [19].

Such feelings, in turn, are connected to negative effects and social impairment [28,29]. They play a role in predicting aggressive behavior and hostile attitudes [30,31];and reduce the capacity of RASI to control negative feelings and actions, even toward innocent people who are not related to the anger-triggering event [20].

### 1.2. Negative Spiral of Hostility and RASI’s Wellbeing

*In one or the other way, this is what breeds violence… to be treated badly by people from here (Spain)… it often leads us to do what we don’t want to do!* These words, spoken by a 37-year-old participant from Colombia, may provide a concrete example of a negative spiraling effect of racial hostility in Western societies.

This effect has been identified in a study by Kamans et al. [24]. Their study was conducted in The Netherlands with 88 young Dutch Moroccans who were aware of the prejudice Dutch people had toward them. These perceptions, in turn, influenced the participants’ tendency to negatively stereotype Dutch individuals. For example, entering a bus and seeing how people immediately held tight to their bags triggered strong negative emotions toward Dutch people, as the young Moroccans saw how their people were generally being perceived as thieves.

These findings highlight that suffering discrimination directly (at a personal level) or indirectly (toward friends or people from the same ethnic group) may result in the development of negative views or behaviors toward the outgroup.

Moreover, these negative views or behaviors may also be triggered by the need to build a strong racial identity against the outgroup, or motivated by anger rumination and a strong desire to reciprocate [32]. Therefore, being exposed to racial discrimination not only damages RASI’s wellbeing; it may also provoke a vicious circle of hostility and anger on both sides, RASI and their host societies.

### 1.3. Discrimination, Loss of Meaning and Potential for Radicalization

Undergoing racial discrimination often leads RASI to resort to a series of unhealthy behaviors that are harmful to themselves and/or to others, such as social isolation, substance abuse, suicidal thoughts [15], joining gangs, or developing extremist views and radicalization [21,22]. In their study, Lyons-Padilla et al. [21] recruited 198 participants with a mean age of 27.42 and found a significant relationships among marginalization, discrimination, loss of significance, and support for extremist groups. Moreover, those authors found that discrimination had a strong negative impact on the individual’s sense of self-worth, a dimension that is particularly important for young people, as they actively strive for self-identity, significance, and acceptance [33].

Findings such as these challenge researchers to develop adequate means to identify groups of RASI who might resort to harmful coping strategies as a way to deal with ethnic hostility in Western countries. Given that millions of RASI have no other option but to live in societies that directly or indirectly treat them in a hostile manner, professionals and academics from different fields are called upon to put forward strategies to prevent or reduce the damage resulting from racism and discrimination [34]. In this regard, Kozan and Blustein [35] highlighted that, practitioners working with disadvantaged populations are challenged to use their clinical experience for advocacy purposes, and thus to contribute to fostering structural change in societies. Mallinckrodt et al. [36] argued that the scientist-practitioner-advocate model is particularly relevant when those served are affected by social justice issues. This model challenges practitioners not only to work with the symptoms presented by individuals who experience racial hostility, but also to work to prevent those experiences by addressing social justice concerns.

As mentioned, anger is the emotion most commonly triggered by racism and discrimination [37]. It can have severe consequences for both the wellbeing of those affected and the social relations in host countries. According to the way anger is cognitively processed (anger rumination), it may have a strong impact on how the affected persons react to an anger-triggering event. Therefore, it is important to explore the degree of anger rumination in RASI undergoing discrimination in Western countries, as a means of implementing strategies to help those who are most affected. Sukhodolsky et al. [26] developed an important instrument, the Anger Rumination Scale (ARS_19), to assess the way individuals cognitively process anger and the length of this emotion once it is triggered by a specific event.

Since the topic of racial hostility and discrimination often brings painful memories and may trigger strong emotions in those affected, a previous study indicated that it could be adequate to find an instrument that uses the Zaltman Metaphor Elicitation Technique [38], which employs imagery or other sensory stimuli to unravel deep-seated emotions while reducing the need for oral language; this is an important advantage when working with RASI living in Western countries.

Although a range of tools exist that assess the impact of discrimination on the wellbeing of RASI [39], we did not find an instrument that identified RASI’s levels of anger rumination that are triggered by real scenarios of racial hostility. Therefore, based on the results from a previous qualitative investigation with RASI and helping professionals [23], our main objective in the present study was to identify the presence of a spiral of anger rumination in RASI as a consequence of experiencing racial hostility in the host country, so that professionals may be better equipped to support RASI while preventing negative effects at individual and social levels.

To achieve this objective, the following actions were identified:Building a new tool, the Picture Elicitation Instrument (PEI) to be used by professionals in Western countries to identify RASI’s potential reactions to real scenarios of discrimination, to facilitate interventions with RASI affected by racial hostility, and to collect data for advocacy against ethnic discrimination;Exploring whether the PEI, along with the ARS_19, can recognize groups of RASI according to their emotional reactions to scenarios of hostility and discrimination;Exploring whether the PEI, along with the ARS_19, can identify categories of RASI on the basis of specific emotions triggered by discrimination scenarios; andChecking whether real scenarios of hostility may influence the degree of social isolation and withdrawal in RASI from different age groups.

## 2. Method

### 2.1. The Picture-Elicitation Instrument (PEI)

The building process for the PEI followed various steps. First, information was gathered from the literature [40] regarding racial discrimination contexts in Western societies and the basic emotions that are frequently triggered in RASI by those scenarios. Another important source of information was a qualitative study conducted by the authors [23] in Madrid (Spain). In that study, 15 semi-structured interviews were conducted with RASI with ten nationalities, along with seven interviews of professionals working with RASI in six NGOs. The results showed that discrimination, in the forms of hostility, mistrust, and anger attitudes or behaviors, was experienced by almost all participants and confirmed by most of the professionals. Basic services, such as house renting, social and health care services, banking, use of public places such as bus stops and public gardens, etc., were all mentioned as examples in which participants felt openly or subtly ostracized due to their ethnic or religious backgrounds. Although that study was conducted in Spain, the literature shows that these contexts of discrimination toward RASI are common in other Western societies as well [41,42,43].

Given these results, some criteria for portraying discrimination contexts were established. One cartoonist drew eight scenarios of discrimination; another sketched six emotions for the answering scale and an image representing social isolation. Various adjustments were made to the drawings based on dialogue between the authors and the two cartoonists.

The final drawings were included in an online survey (using Google Forms) and sent to experts who provide support for RASI in several countries. The experts provided input on the eight scenarios and the six emotions. Two important criteria for selecting the experts were their amount of experience in working with RASI, and that they were from as many different countries as possible, so that the new PEI instrument would be relevant to various cultures. We conducted a pilot study to assess whether there was a need to make adjustments in the drawings or other aspects of the PEI before sending the material to a larger number of professionals and academics. The stepwise building process of the PEI is shown in Figure 1.

As shown in Figure 1, 34 professionals and academics with 22 nationalities, working in 18 countries from four continents, provided their input to the PEI. Around 46% of these professionals had been working with RASI for more than five years; the majority were psychiatrists, psychologists, and social workers. Several were working in very difficult contexts, such as those in Afghanistan, Lebanon, or Malta. The goal of the process was to build a tool that was as culturally sensitive as possible, to be used by professionals working with RASI to explore the negative impact of racial hostility, facilitate interventions to reduce this impact on those affected, and advocate against society’s overt or subtle mistreatment of RASI by using the PEI instrument for data collection on discrimination. In this way, the new PEI tool would encourage professionals working with RASI to use the scientist-practitioner-advocate model [36].

It is important to highlight that the PEI is not proposed as a tool to rigorously measure the degree of psychological impact of discrimination on the wellbeing of RASI; instead, it is put forward as an instrument to help professionals plan and evaluate interventions with this population, to advocate against social hostility toward RASI, and to promote positive intergroup relations.

As shown in Figure 1, the final instrument includes six negative and two positive scenarios. The six negative illustrations were based on racist situations experienced by RASI living in Madrid, whereas the two positive illustrations were based on suggestions provided by participants to promote the integration of RASI into Western countries [23]. One scenario of the PEI is presented in Figure 2.

The PEI has the format of an A5 booklet; each scenario is depicted in a left page with the respective answering scale on the right-side page, so that participants see and respond to only one scenario at a time. The emotional answering scale includes drawings of four basic emotions (joy, sadness, anger, and despair), as shown in Figure 3. The emotion “joy” was included mostly in the two positive scenarios.

The multi-scenario (Figure 4a) consists of the six negative scenarios placed together on the left-side page of the booklet, while on the right-side page a drawing illustrating withdrawal, loneliness, sadness, and powerlessness appears. This sketch is shown in Figure 4b. The question for the multi-scenario is: *As you look at the above situations (six discrimination scenarios) and remember the times this happened to you or to your friends, please say how much you feel like taking this behavior* (ranging from 0—never, to 10—always). This question was motivated by two previous findings: the existence of a relationship between racial discrimination and social isolation [44], and the fact that experiencing ethnic discrimination toward oneself or toward family members or friends can have a similar impact on a person’s wellbeing [45,46].

After building the PEI, a quantitative study was conducted to explore a potential spiral of anger rumination in RASI living in Madrid (Spain) as a consequence of experiencing racism and discrimination. The study included the PEI, the ARS_19, and a short sociodemographic questionnaire.

#### 2.1.1. The Anger Rumination Scale_19 (ARS_19)

The ARS_19 seemed to be an appropriate instrument for measuring RASI rumination on anger, since this emotion is the one that is most frequently triggered by racial hostility and discrimination [37,47].

The scale was developed to assess cognitive processes activated by anger-provoking situations [26], and it measures the tendency to ruminate on present anger events and to recall past anger experiences. The instrument has four subscales: “angry memories” (continuously thinking about past anger-triggering events), “angry afterthoughts” (the tendency to think again and again about recent anger events), “thoughts of revenge” (the tendency to get back at the person or persons who provoked the anger situation), and “understanding the causes” (having thoughts about the reasons underlying the anger situations).

The ARS_19 has been tested and validated in different countries and translated into different languages [48,49,50,51,52,53], showing solid psychometric properties in all those studies.

#### 2.1.2. Sociodemographic Questionnaire

Along with the PEI and the ARS_19, a short sociodemographic questionnaire was included in the study, with questions related to age, sex, nationality, year of arrival in Spain, legal status, and experiences or perceptions of mistrust by the host society toward one’s self or significant others.

### 2.2. Procedures

#### 2.2.1. Data Gathering

It must be stated that it is very difficult to obtain a representative sample of RASI in Spain, since many of them are irregular residents (i.e., they have no residence permit); thus, many of them do not access social services from public institutions. For this reason, the RASI sample was obtained from six NGOs in the city of Madrid. The authors have no affiliation with any of the NGOs, nor are they working in Madrid. Permission was given by the directors of each of the NGOs, to whom the detailed project was sent before the start of data gathering.

Those six NGOs were determined on the basis of a previous qualitative study conducted by the authors [23]. The six NGOs have the following characteristics: (1) they serve large numbers of RASI in Spain, regardless of their legal status; (2) they provide assistance to RASI in a great diversity of situations involving many nationalities and cultures; and (3) their work is recognized at both national and international levels.

Before applying the questionnaire, participants were given all information about the project, were ensured that confidentiality would be maintained at all times, and were asked for oral and written informed consent. After the interviews, time for debriefing was offered, according to individual needs. It is worth noting that all data were gathered by the first author, who is a clinical psychologist with wide experience in providing psychological support to migrants and refugees.

#### 2.2.2. Sample

The present sample included 509 RASI from 15 Latin American countries, with ages ranging from 18 to 65 years (M = 35.36, SD = 11.06). The countries with the greatest representation were Honduras (28%), Peru (17%), Nicaragua (12%), and Colombia (11%). The majority of the participants were women (70.3%) between 18 and 39 years of age (66.6%) who had been in Spain for less than two years (69.9%). Regarding legal status, 74% of the participants had no residence permit (i.e., they were irregular immigrants or failed asylum seekers), 13% were asylum seekers or refugees, and 13% had a residence permit. Although there was no intention of limiting the sample to Latino people, Latinos form the majority of RASI who are currently searching for services from NGOs in Madrid [54].

## 3. Results

To reach the main goal of this study—to “identify the presence of a spiral of anger rumination in RASI, as a consequence of experiencing racial hostility in the host country” —we proposed four objectives. The first objective—“to build a new tool that could be used to explore RASI’s reactions to scenarios of racial hostility, facilitate interventions with these population, and collect data for advocacy purposes”—was achieved through the process of constructing the instrument with the input of international experts. 

To address the second objective— to “explore how the PEI with the ARS_19 can identify categories of RASI according to their emotional reactions to real scenarios of discrimination”—multivariate techniques were used (factor and cluster analyses), aided by SPSS software, version 24. Principal component analysis was conducted to assess the four factors of the scale: “angry memories”, “angry afterthoughts”, “thoughts of revenge”, and “understanding the causes”, which were developed by the authors [26]. Six items (see Table 1) did not load onto the corresponding factors; nevertheless, the remaining 13 items loaded perfectly onto the four factors of the ARS_19, with the model explaining 61.6% of the variance for the total set of variables.

The ARS_13 revealed strong internal consistency for the whole scale (α = 0.84), and moderate-to-good internal consistency for the four subscales: “angry memories” (α = 0.77), “angry afterthoughts” (α = 0.67), “thoughts of revenge” (α = 0.66), and “understanding the causes” (α = 0.60).

After these analyses, a *K*-means cluster analysis was conducted to enquire whether groups of participants could be categorized using the four factors of the ARS_13 scale. Clatworthy et al. [55] emphasized that cluster analysis offers a major contribution to various health sciences, as it helps to identify groups that may profit from specific interventions. The analysis classified the 509 participants into four categories, according to the ways they processed anger triggered by the discrimination scenarios of the PEI. The four groups were named and are explained in Figure 5. A further analysis was made to enquire whether there was a difference between sex/gender in the four clusters; however, no significant difference was found.

After these two analyses, a multiple correspondence analysis (MCA) was used to check the study’s third objective—to “explore whether the PEI, along with the ARS_19 can identify groups of RASI, on the bases of specific emotions triggered by discrimination scenarios”. This analysis examined whether the four categories (“social desirability”, “chewing”, “grudge”, and “vengeful”) were related to the emotions of the PEI (“sadness”, “anger”, “despair”, and “joy”) that were chosen by participants as they observed the six discrimination scenarios. It is worth highlighting that, as mentioned in Section 2.1, the emotion “joy” was included most in the two positive scenarios. The results of the MCA are presented in Figure 6.

As shown in the MCA plot, the two dimensions explained 59.5% of the variance in the data. The top part of the graph presents two different emotions, one more passive (“sadness”), associated with those with a tendency for “social desirability”, and one more active (“anger”), closely associated with those RASI with a tendency for “chewing”. On the lower part of the graph appear the other two categories, “grudge” and “vengeful”, which also seem to be more active. These two groups of RASI highlighted “despair” as the most common emotion triggered by the six discrimination scenarios.

After this analysis, a further step was taken to address the fourth objective—“to check whether real scenarios of racial hostility may influence the degree of social isolation and withdrawal in RASI from different age groups”. A second MCA was conducted to investigate how the four groups (“social desirability”, “chewing”, “grudge”, and “vengeful”) behaved with respect to age groups (in four categories: 18–29; 30–39; 40–49; and 50 and above) and levels of withdrawal (in three categories: 0–4; 5–7; and 8–10). The two dimensions explained 82.1% of the variance in the analysis, as shown in Figure 7. 

As can be observed, participants from the two categories vengeful” and “grudge” continued to behave in a similar manner. Moreover, those RASI who most identified with those two categories were the youngest (18–29 years), who also showed the strongest tendency for social isolation in the multi-scenario (8–10). The “chewing” category was formed by two age groups: people relatively young (30–39 years), and participants older than 50. Those two groups had middle scores (5–7) in the multi-scenario. Finally, the “social desirability” category was formed by middle-aged RASI (40–49 years) who were at the lower end (0–4) in the multi-scenario, showing a tendency to react to hostility and discrimination in more proactive ways, rather than isolating themselves. These results are discussed in the next session.

## 4. Discussion

As mentioned above, the present work had a major goal: “to identify the presence of a spiral of anger rumination in RASI, as a consequence of experiencing racial hostility in the host country, so that professionals may be better equipped to support RASI, while preventing negative effects at individual and social levels”. The aim of this article was not to explore the resilience of the RASI; in a previous stage of the investigation, resilience or coping factors were explored by the authors [23].

To achieve our goal, four objectives were established. The first objective was to build a new tool that was as culturally sensitive as possible, which could be used by professionals to identify RASI’s potential reactions triggered by real contexts of racial hostility; to facilitate interventions with RASI affected by discrimination; and to enable data collection for advocacy purposes. Therefore, by using the PEI, helping professionals may be stimulated to engage in the scientist-practitioner-advocate model [36] and thus to provide better services to RASI, while actively contributing in improving social relations in host countries. This objective was met through the construction of the new tool.

The other three objectives were to explore whether the PEI, along with the ARS_19, could identify the following: categories of RASI, according to their emotional reaction to discrimination scenarios (objective 2); groups of RASI, on the basis of specific emotions triggered by racial hostility scenarios (objective 3); and whether real scenarios of discrimination influence the degree of social isolation and withdrawal in RASI from different age groups (objective 4).

The results of the present study offer important insights into the second, third, and fourth goals of the investigation, and thus seem to contribute positively to the literature on the topic of social relations between RASI and host societies. First, the cluster analysis showed four ways in which RASI may react to anger-triggering events connected with discrimination, namely, “social desirability”,” chewing”, “grudge”, and “vengeful”.

Likewise, the two MCA conducted on the data helped in understanding how the contexts of racial hostility presented in the PEI may influence social isolation levels in different age groups of RASI. Although somewhat promising, these results need to be viewed with caution, considering that this is the first study using the ARS_19 with the PEI for RASI. Moreover, as described in the results, the ARS_19 was tentatively adjusted in the present study. Therefore, academics and practitioners are encouraged to conduct further studies to validate the instruments with RASI populations and, simultaneously, to evaluate the present findings.

Before reflecting further on the main findings, it is important to discuss the possible reasons for adjusting the ARS_19 in our sample.

### 4.1. Measuring Anger Rumination in RASI: Controversies in Regard to ARS_19

The ARS_19 was selected for the present investigation because of its strong validity and reliability in assessing cognitive patterns following anger-triggering episodes. However, in our sample, statistical analyses did not support the full version of the ARS_19. The present results could be explained by at least two important factors: the nature of the sample, in comparison with samples in previous studies, and the contextual factors related to RASI’s experiences before, during, and after migration.

#### 4.1.1. Nature of the Sample

Regarding differences in the sample, it is important to highlight that the numerous studies found in the literature [49,50,51,52,53] that used the ARS_19 to assess anger rumination levels were conducted with homogeneous populations, mostly university students living in their own countries. Our sample, however, was formed by RASI living in a country other than their own who had often experienced violence, war, persecution, or extreme poverty.

Moreover, in the samples from previous studies, participants seemed to have some control over their lives. In our sample, however, RASI’s decision to migrate was not a voluntary choice, but rather a consequence of political conflicts, indiscriminate violence, economic hardships, and other perils. During the process of data gathering, a large number of participants mentioned that if the situation were better at home, they would not have left their country.

#### 4.1.2. Contextual Factors: RASI’s Experiences before, during, and after Migration

The present performance of the ARS_19 could also be related to contextual factors of the sample. Tuomisto and Roche [56] highlighted that although anger is considered to be a universal emotion with common affective and cognitive components, it is also the emotion most affected by contextual factors. Therefore, anger may be differently felt and processed in RASI’s countries of origin, compared to the way it is felt and processed by RASI living in Western countries.

The majority of RASI in Western countries are aware of the hostile contexts in which they live, as these are often permeated by ethnic discrimination, social mistrust, labor exploitation, future uncertainty, fear of deportation, and other factors. These stressful situations greatly affect RASI’s wellbeing, especially with respect to their mental and physical health [11,17,57,58]. Therefore, the way anger is cognitively processed by RASI is probably different from the way it would be processed if they were living in their native countries, where they would know their rights and be familiar with adequate ways of handling anger-provoking events. Some of these reasons might also explain the nonsignificant differences between men and women in each of the four clusters of the ARS_13. In fact, results from previous studies were somewhat contradictory, with some authors finding sex differences in one or two factors of the ARS, while other authors did not find significant differences [26].

Having this in mind, the following sections deal with the performance of the ARS_13 with the PEI in adhering to the objectives of the present study.

### 4.2. Categories of RASI and Emotional Reactions to Discrimination Scenarios

The results are organized and discussed in detail in three following sections, considering the four categories of RASI (“social desirability”, “chewing”, “grudge”, and “vengeful”) and the way they relate to the four age groups (18–29, 30–39, 40–49, and 50 or over), the four emotions of the PEI (“sadness”, “joy”, “anger”, and “despair”), and the three levels of social isolation (0–4, 5–7, and 8–10).

#### 4.2.1. “Social Desirability”: Middle-Age, Sadness, and Low-Level Withdrawal

This category included middle-aged adults (39–49) who tended to deal with anger rumination in a socially desirable manner, rather than in a resentful or retributive way. Moreover, in this group the most common emotion triggered by discrimination scenarios was “sadness”, although in the multi-scenario they gave low scores (0 to 4) to social isolation tendency. In other words, despite being emotionally affected and sad due to discrimination, this group refused to capitulate in the face of ethnic hostility and mistreatment.

This result is somewhat contradictory, since it is well known from previous findings that feelings of sadness often lead to social withdrawal, and that both sadness and social withdrawal are important predictors of poor mental wellbeing in RASI [17]. The two indicators of wellbeing do not go together in our sample, meaning that despite feeling emotional pain due to discrimination and hostility, along with other post-migration stressors, these RASI seem to fight against sadness to reach their goals.

This could be explained by two factors: first, the somewhat proactive behavior that could result from a certain sense of control over personal life which, according to Jang, Chiriboga, and Small [59], can be a protective factor against the damaging effects of racial discrimination and even help to maintain resilience levels in adversities.

Second, RASI in this group may focus on sadness in a reflective pondering way, as opposed to giving in to a passive or gloomy attitude. Treynor et al. [60] conducted a study on sadness rumination and found a significant difference between these two ways of dealing with sadness. Reflective pondering implies that the person engages in cognitive problem-solving to alleviate depressive tendencies. This attitude contrasts with brooding, which is a passive attitude that leads to an increase in depressive symptoms. In our sample, the “social desirability” group probably felt sadness in the context of discrimination and hostility, but other factors were stronger in leading them to engage in reflective pondering as a means to solve problems and achieve their goals. The goals of fighting for a job and raising the family may lead RASI to focus less on anger rumination, while concentrating more on undertaking adaptative, positive, and proactive behaviors [61].

#### 4.2.2. “Chewing”: Young and Old, Anger, Mid-Level Withdrawal

The “chewing” category, composed of RASI who scored high in short-term anger memories, included participants of two different age groups: 29–39 and 50 or more. These two groups also had similar scores in the multi-scenario question, providing numbers between 5 and 7 on the tendency to social isolation as a result of racial hostility. It is surprising that although life experiences in these two age groups are surely very different, both groups showed similarities in both anger rumination resulting from discrimination, and withdrawal tendencies as they witnessed hostility scenarios. Once again, these results somewhat contradict previous findings [62], which emphasized that older adults are less prone than younger adults to dwell on anger experiences.

According to Schieman [63], contextual situations and socio-structural and cultural environments are experienced differently by younger adults when compared with older adults, and this fact influences anger outcomes. Consequently, personal gains and losses that are associated with different ways of dealing with them may influence anger processes in different age groups. Therefore, the contextual circumstances lived by RASI in the younger group (29–39) may have led them to evaluate discrimination and hostility experiences as barriers to achieving important goals in the host country, such as finding new relationships, having a career, or finding a good job. Moreover, the sense of being treated unjustly on the basis of ethnicity may trigger stronger anger in the younger (29–39) RASI than in the older RASI (50 or more) [59].

Having reflected on these points, however, it is important to remember that regardless of their age, the priority for RASI in Western countries is usually to find a job to cover their basic needs and those of their families. This priority, in turn, becomes more difficult as age increases. For example, being a RASI over 50 usually means having much fewere opportunities for work, while strongly feeling the responsibility to care for elderly parents or dependent sons and daughters in the country of origin. Together, these factors may lead to feelings of powerlessness and frustration that may turn into anger, especially as RASI face experiences of racial hostility added to age discrimination in their search for work in host countries. As Berkowitz [64] asserted, frustration, sadness, and depression are negative emotional states that often give rise to anger and aggression. This could be the case with RASI over 50, who probably feel torn between the urgency to meet basic needs and the sense of injustice experienced when hostility, racist acts, or racist comments are directed toward them due to their origin or older age.

#### 4.2.3. “Vengeful” and “Grudge”: Youngest, Desperate, High Withdrawal

Those in the categories of “vengeful” and “grudge” were, simultaneously, the youngest in the sample (18–29) and those expressing greater tendency for social isolation and “despair” as they witnessed scenarios of discrimination and hostility. This could be related to young people having more dreams and hopes for a bright future and, simultaneously, a stronger sense of social justice and fairness [61,65]. Both of these feelings are under threat as young RASI experience racial hostility and encounter numerous barriers to achieve their goals in the host country.

Being young, new to the country, unable to achieve personal goals, having no documents, and feeling powerless in fighting against racial injustice may be among the reasons that lead this group to highlight “despair” as the most common emotional state. Associated with despair comes the tendency for social isolation and withdrawal, a characteristic that is strongly connected with anger rumination [29] and often with aggressive behavior [66].

Deffenbacher [61,65] argued that provocative events, disrespectful and unjust treatment from others, seeing one’s values being violated, and being stopped from reaching one’s deserved goals can all lead to strong feelings of anger. These feelings tend to be even more intense when others’ offensive actions or behaviors are assessed as intentional, malevolent, preventable, and/or deserving punishment. The literature further shows that negative treatment by other people, such as being rejected, ostracized, or discriminated against, triggers feelings of insecurity and threatens the individual’s need for basic trust and acceptance by others [61,67]. Such hostile treatment could lead this young group of RASI to experience sadness, anger, powerlessness, or hurt, since these feelings are often connected with a threat to one’s sense of identity and self-worth [68].

Although the need for acceptance and belonging is important for every human being, it is even stronger for younger people, as the beginning of adulthood is an important life stage for the development of cultural and social identity [33]. For this reason, perceived threats to self-worth, such as those experienced in ethnic discrimination and hostility scenarios, could be another reason leading the youngest group to highlight “despair” as the most common feeling. The lack of a cultural identity and a sense of belonging, often damaged by racism, discrimination, and hostility, has been emphasized by Lyons-Padilla et al. [21] as a potential trigger for seeking support from extreme groups who are often seen as a source of meaning and purpose. Added to this, the youngest group showed the highest tendency for withdrawal and isolation (8–10), a result somewhat in line with the literature.

According to Alcalá et al. [69], young people are particularly vulnerable to discrimination and hostility, which often leads to social isolation; this, in turn, may strongly influence radicalization. Therefore, the present results not only reveal the harmful consequences of ethnic discrimination that, per se, strongly affects the psychosocial wellbeing of young RASI [59], but also highlight the need to develop and implement strategies to reduce young RASI’s suffering, to counteract their tendencies for isolation, despair, and anger, and to avoid harmful consequences for them and for Western societies as a whole.

## 5. Limitations

At least three limitations can be identified in the present work. First, all RASI in the sample were from Latin American countries. Although this is somewhat in line with the wave of Latinos arriving lately in Spain, the results may not be representative of RASI from other nationalities living in Spain. Therefore, conducting this study with more heterogeneous samples regarding nationality would surely make the results more applicable to the wider RASI population.

Second, it would have been helpful to deeply consider sex differences from the outset of the study. This was not done because the study was already quite comprehensive, and our major objective was to build the PEI and explore the impact of racial hostility on the emotional wellbeing of RASI. In further studies, it would be important to include the variable of sex/gender from the outset of the study to enquire how the PEI, along with the ARS_13, performs with the two groups of RASI.

Third, the method of applying the questionnaire could have triggered social desirability issues. Very often, participants would look first at the anger emotion but end up choosing the sadness one. Previous research [70,71] argues that discriminated-against groups often have a tendency to under-report experiences of racial hostility, especially when racism is internalized as something normal in host societies. It would, therefore, be useful to conduct further studies with these two instruments using a computer or a smartphone app to prevent social desirability issues. This is being planned by the authors through a study using the two instruments with larger groups of RASI living in different countries.

## 6. Suggestions for Professionals and Academics

We consider that the present results are a modest contribution for professionals and academics working or doing research with RASI. For example, the six negative scenarios of the PEI may help as a door-opener for professional interventions with RASI, as they enable (in a non-intrusive way) the sharing of painful experiences and feelings provoked by racial discrimination. In fact, in the process of data gathering, we found that the scenarios of the PEI were a valuable tool in helping participants connect with painful emotions (many wept as they looked at some scenarios and spoke about their experiences). Such emotions are often repressed by a need for survival in host countries. In a similar manner, the multi-scenario may also be a good measure for use by professionals in identifying RASI who socially isolate as a result of racial hostility, and thus in developing and implementing strategies to counteract this behavior.

Moreover, helping professionals and academics in the use the PEI to gather data on discrimination experiences lived by RASI can have two purposes: developing advocacy strategies, and contributing to the further testing of the PEI, together with the ARS_13 while putting forward new suggestions for their use.

In short, we think that these results may be a source of encouragement for professionals to grow in the use of the scientist-practitioner-advocate model by developing interventions for working with affected RASI and gathering data for advocacy actions in defense of their human rights. It is worth emphasizing that helping professionals and academics who work with RASI are in a privileged position to make relevant contributions to the development of more compassionate societies, in which all social actors are respected and welcomed, regardless of their places of birth.

## Figures and Tables

**Figure 1 behavsci-12-00180-f001:**
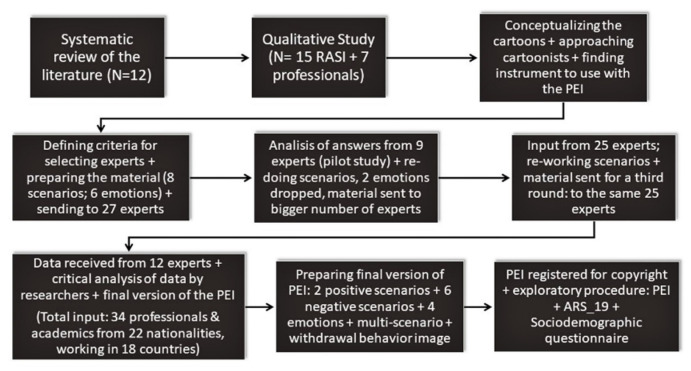
The PEI stepwise building process.

**Figure 2 behavsci-12-00180-f002:**
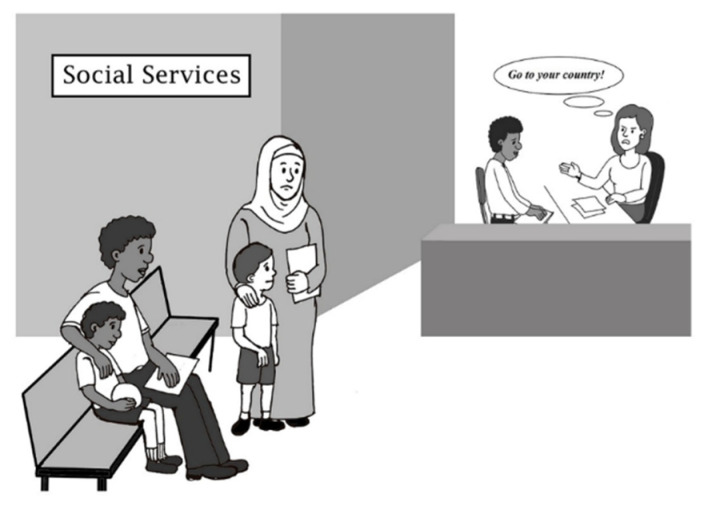
Scenario 8 of the PEI.

**Figure 3 behavsci-12-00180-f003:**
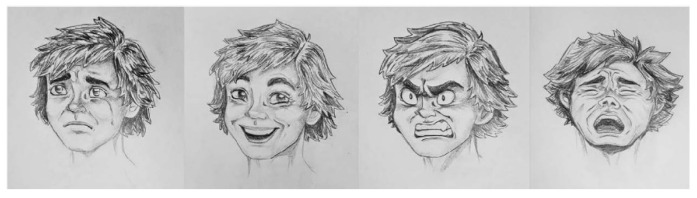
Emotional Scale of the PEI.

**Figure 4 behavsci-12-00180-f004:**
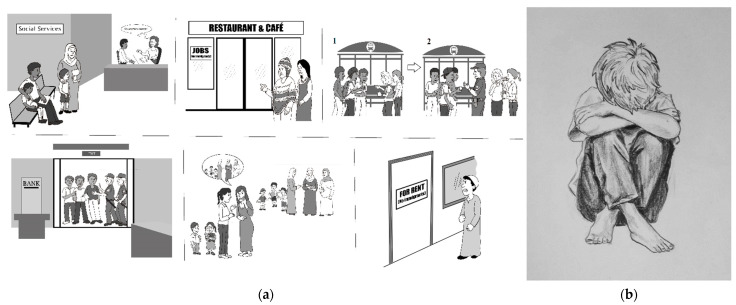
Multi-scenario of the PEI and withdrawal behavior. (**a**) Multi-scenario of the PEI; (**b**) Withdrawal behavior shown in the PEI.

**Figure 5 behavsci-12-00180-f005:**
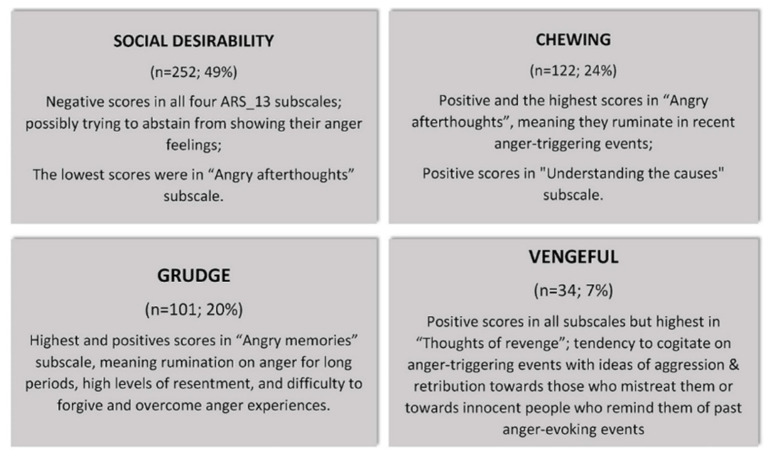
Cluster analysis: description of the four categories.

**Figure 6 behavsci-12-00180-f006:**
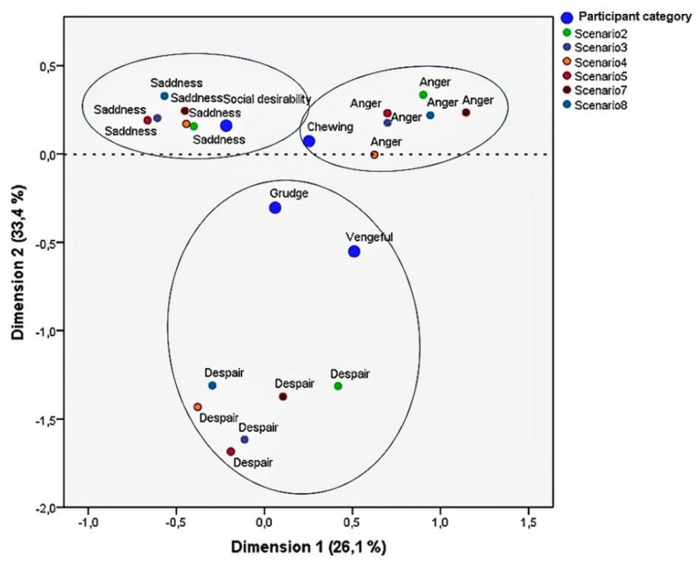
Multiple correspondence analysis map: ARS_13 and the PEI emotional scale.

**Figure 7 behavsci-12-00180-f007:**
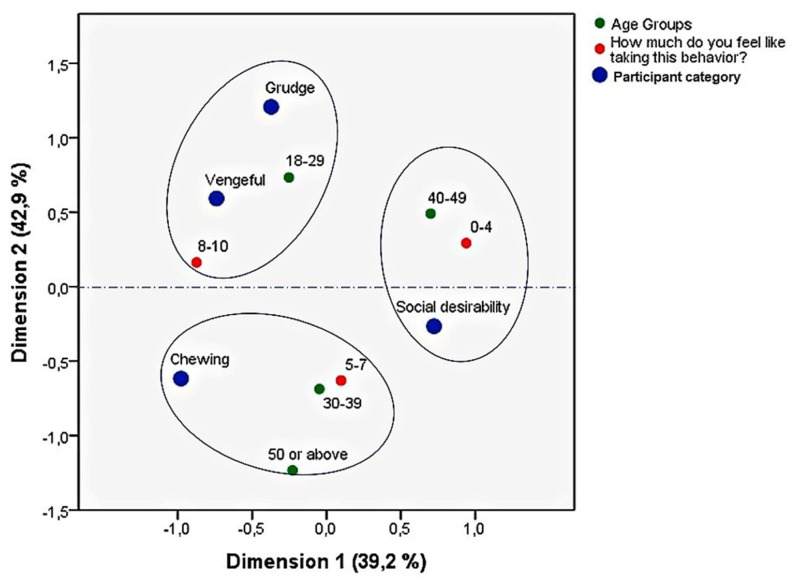
Multiple correspondence analysis map: ARS_13 Age groups and withdrawal levels.

**Table 1 behavsci-12-00180-t001:** Items withdrawn from ARS_19, according to the four factors.

Factors	Items
*Angry memories*	2. I ponder about the injustices that have been done to me
	14. I feel angry about certain things in my life
*Angry afterthoughts*	7. After an argument is over, I keep fighting with this person in my imagination
	8. Memories of being aggravated pop up into my mind before I fall asleep
*Thoughts of revenge*	13. I have day dreams and fantasies of violent nature
*Understanding the causes*	10. I have had times when I could not stop being preoccupied with a particular conflict data

## Data Availability

Data and study materials are available upon request to the authors.

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
