# Peer review of "Chewing Revenge or Becoming Socially Desirable? Anger Rumination in Refugees and Immigrants Experiencing Racial Hostility: Latin-Americans in Spain"

_behavsci, 2022, doi:10.3390/bs12060180_

Round 1

Reviewer 1 Report

The authors provide a well structured and detailed article.
The theme of the research is innovative enough as to be of interest for its publication.

However, I would suggest the authors to include at least a few more references of related works from outside Spain, in order to contextualise the living condition of RASI around the world, including for the identity struggle and social hostility.
I would suggest, for the Italian case, reading the article titled:

Bianco; Ortiz. “Soy solo un refugiado”: inferioridad y temporalidad en las identidades en migración.

Author Response

Dear Reviewer:

We are grateful for your suggestions, which have allowed us to improve the quality of our article.

You can see the responses to each of the questions in the attached document.

Yours sincerely,

The Authors

Reviewer 2 Report

The present work deals with the worrisome scenario of racial hostility in host countries using the Spanish scenario of the most recent years to address the investigation. The sample population was recruited in NGO and was composed of Latinos, but despite not representing other RASI nationalities, the value of the work is to establish a first step based on a homogeneous cultural population (the authors discuss it as one of the limitations). The work is aimed to build a tool (PEI) that, combined with ARS_13 (ARS_19 had 6 items not suitable) would allow identifying a spiral of ‘anger rumination’ in RASI that results from experiencing racial hostility from host countries. A cluster analysis provided 4 clearly distinguished reactions to anger-triggering events related to discrimination. A final aim was also to offer a tool also to provide them support and preventing negative effects. The work is very well written and sustained by the literature. The authors indicate the need to validate the instruments with the RASI population while replicating their findings. They also discuss about controversies around ARS_19 (adjustement to ARS_13, etc), and different aspects of the four categories.

Some aspects still need attention and to be addressed by the authors. The main one is the gender perspective (which is lacking); there’s no reference to resilience (essential to be considered if one of the final aims is to provide support), some racial aspects of the PEI and the title.

  • Sex/gender perspective is not clear and should be stated from the beginning (introduction) as it is a key aspect. Depending on the country, the gender expected is different. On the other hand, the psychological constructs/psychiatric profiles differ between sexes/genders.

    Also, surprisingly, most participants recruited were women (70.3%), probably because where NGO, or for other reasons. However, the analysis of the different key issues differs in women/men (if we do a dichotomic, binary) and non-binary people. Please, discuss.

Were the psychometric properties of ARS_13 similar/different between men/women RASI?

  • Aspects on resilience should be presented and discussed, as they are also playing a role in these scenarios. They are indirectly mentioned in lines 214-215 in the form of positive scenarios to promote integration.

  • Figure 2, illustrates real settings, with people of different nationalities easily identified by their appearance and spatial distribution. However, in the definition of PEI, the authors say (line 167) The building process of the PEI followed various steps. Firstly, information was 167 gathered from the literature [36] regarding racial discrimination contexts in Western soci-168 eties, and the basic emotions frequently triggered in RASI by those scenarios.

So, one would expect that figure 3 presenting 4 the ‘emotional scale of the PEI’ in a 4 Likert scale would illustrate ‘the basic emotions triggered in RASI’ not in a western character. The same applies to Figure 4 (b) since it seems to be the same western appearance boy of figure 3. Also, the age is of a child (more than even a teenager), so to which extend it fits with a real subject where ‘anger reaction’s are expected / studied. The work is presented in terms of adults.

  • While the authors discuss, present the limitations and talk about the ‘extend’ of the current contribution, in a clear exercise of scientific responsibility, considering it is a new tool and the clustering is based on a specific sample (Latinos) in a specific western country (Spain) (where Latino nationalities are ‘less discriminated’ that others with arabic origen, for instance), the title overestates the generalization that, at this stage of the progress with the tool, can be done. It should be informative of the findings, and the sample used and the host country studied. As the title states right now it could be just a review on these facts and actors.

Other minor points are:

Line 30. RASI’s should also be defined in the introduction when it appears for the first time

Line 44. Please, check ‘560 thousand third country nationals’

Line 48. Reference [13] should go at the end of the sentence as it is a citation of this HRW report.

Line 86. Add the reference with number.

Table 1. Please, add headings to each column (Factors)(items)

A graphical abstract on the conceptual frames (anger rumination, factors, etc) is strongly recommended and it would be very useful for several other purposes.

Author Response

Dear Reviewer:

We are grateful for your suggestions, which have allowed us to improve the quality of our article and to identify some limitations that may lead to future lines of research.

You can see the responses to each of the in the attached document.

Yours sincerely,

The Authors

Round 2

Reviewer 2 Report

The authors have provided rationals for the questions arised and have included the suggestions in their Ms.
Just a couple of things are pending:

With regard to:
2) 2             Aspects on resilience should be presented and discussed, as they are also playing a role in these scenarios. They are indirectly mentioned in lines 214-215 in the form of positive scenarios to promote integration.

The objective of article was not to explore the resilience of the RASI; in a previous stage of the investigation, resilience or coping factors were explored and are published in the Journal of Social Work (2021).

In the study presented here, the positive scenarios were highlighted as importnat ways to promote the integration of RASI in host countries.

RE: Please, include this consideration in the discussion. It is important and it is noteworthy for readers to be able to find the previous work published on 2021.
--

readers to be able to find the previous work published on 2021.

3             Figure 2, illustrates real settings, with people of different nationalities easily identified by their appearance and spatial distribution. However, in the definition of PEI, the authors say (line 167) The building process of the PEI followed various steps. Firstly, information was 167 gathered from the literature [36] regarding racial discrimination contexts in Western soci-168 eties, and the basic emotions frequently triggered in RASI by those scenarios.

So, one would expect that figure 3 presenting 4 the ‘emotional scale of the PEI’ in a 4 Likert scale would illustrate ‘the basic emotions triggered in RASI’ not in a western character.

The same applies to Figure 4 (b) since it seems to be the same western appearance boy of figure 3. Also, the age is of a child (more than even a teenager), so to which extend it fits with a real subject where ‘anger reaction’s are expected / studied. The work is presented in terms of adults.

In the building process of the PEI, the "child like" character was used because we did not want to mark the gender features too much.

Moreover, after the discrimination scenarios were painted, and also the emotions, all this material was sent to professionals and accademics in three rounds.

The feedback received from the professionals was very positive regarding both, the discrimination scenarios and the 4 emotions (including the character).

As it can be seen in Figure 1, the 34 accademics and professionals who participated in the rounds were from 22 nationalities and were working in 18 countries. Moreover, 46% of these professionals had been working with RASI for more than 5 years; therefore, we were able to gather very valuable information in the building process of the PEI. 

RE: After this clear argument, then, it will be fine for me.
----

While the authors discuss, present the limitations and talk about the ‘extend’ of the current contribution, in a clear exercise of scientific responsibility, considering it is a new tool and the clustering is based on a specific sample (Latinos) in a specific western country (Spain) (where Latino nationalities are ‘less discriminated’ that others with arabic origen, for instance), the title overestates the generalization that, at this stage of the progress with the tool, can be done. It should be informative of the findings, and the sample used and the host country studied. As the title states right now it could be just a review on these facts and actors.

Please, see below our proposal of new title

Chewing Revenge or Becoming Socially Desirable? Anger Rumination in Refugees and Immigrants Experiencing Racial Hostility: Latin-Americans in Spain

RE:  This is a good proposal, indeed. More specific and useful to highlight your findings.

---

A graphical abstract on the conceptual frames (anger rumination, factors, etc) is strongly recommended and it would be very useful for several other purposes.

Many thanks for this helpful suggestion. Please, see the graphical abstract in next page.

RE: Good work. I really puts your work in more value as it offer a whole idea of the project and scenarios. I’d just suggest to use the palette of different colours for boxes and circles, so it is more colourful

Author Response

Dear reviewer, 

Please find our answers in green.

Best regads,

The Authors

2) 2             Aspects on resilience should be presented and discussed, as they are also playing a role in these scenarios. They are indirectly mentioned in lines 214-215 in the form of positive scenarios to promote integration.

The objective of article was not to explore the resilience of the RASI; in a previous stage of the investigation, resilience or coping factors were explored and are published in the Journal of Social Work (2021).

In the study presented here, the positive scenarios were highlighted as importnat ways to promote the integration of RASI in host countries.

RE: Please, include this consideration in the discussion. It is important and it is noteworthy for readers to be able to find the previous work published on 2021.

RE: Please, find this included in the document, discussion section, lines 382-384

A graphical abstract on the conceptual frames (anger rumination, factors, etc) is strongly recommended and it would be very useful for several other purposes.

Many thanks for this helpful suggestion. Please, see the graphical abstract in next page.

RE: Good work. I really puts your work in more value as it offer a whole idea of the project and scenarios. I’d just suggest to use the palette of different colours for boxes and circles, so it is more colourful

PLease find the new version in a separate document.

--
